# Sub-8 nm networked cage nanofilm with tunable nanofluidic channels for adaptive sieving

Si-Hua Liu[1,4], Jun-Hao Zhou [1,4], Chunrui Wu[2], Peng Zhang[3], Xingzhong Cao [3] & Jian-Ke Sun [1] ✉

Biological cell membrane featuring smart mass-transport channels and sub-10 nm thickness was viewed as the benchmark inspiring the design of separation membranes; however, constructing highly connective and adaptive pore channels over large-area membranes less than 10 nm in thickness is still a huge challenge. Here, we report the design and fabrication of sub-8 nm networked cage nanofilms that comprise of tunable, responsive organic cage-based water channels via a free-interface-confined self-assembly and cross-linking strategy. These cage-bearing composite membranes display outstanding water permeability at the $10^{-5}$ cm$^2$ s$^{-1}$ scale, which is 1–2 orders of magnitude higher than that of traditional polymeric membranes. Furthermore, the channel microenvironments including hydrophilicity and steric hindrance can be manipulated by a simple anion exchange strategy. In particular, through ionically associating light-responsive anions to cage windows, such 'smart' membrane can even perform graded molecular sieving. The emergence of these networked cage-nanofilms provides an avenue for developing bio-inspired ultrathin membranes toward smart separation.

In nature, cell membranes are selectively permeable and can adaptively transport water and other nutrients to maintain cell viability, owing to the fantastical architecture with smart and highly selective protein channels (e.g., aquaporin 1) aligned in the ultrathin lipid bilayer (less than 10 nm in thickness)[1–3]. Crafting artificial membranes with similar adaptive channels and functions has profound implications for practical applications[4]. Emerging artificial channel materials, such as 1D carbon nanotubes[5], 2D nanosheets (e.g., graphene and conjugated-polymer-framework)[6–8], and 3D crystalline frameworks (e.g., metal-organic frameworks and covalent organic frameworks)[9,10], were reported to possess well-defined nanofluidic channels with mass-transport behaviors comparable to their biological counterparts. The major downside of these channel materials is their poor processability

caused by the inherited insolubility, i.e., the high surface energy of the nanodispersions makes them prone to aggregate in solution, thus leading to nonselective interface defects especially in large-scale membrane synthesis. In addition, it remains technically challenging to control the channel length and align the channels parallel to the membrane's cross-section to reach a minimum mass-transfer path[4,11,12].

Porous organic cages (POCs) as a type of intriguing 0D nanoporous molecule possess salient features of easy-to-modify skeletons and good solubility, which offer options for physicochemical structure regulation and scalable solution-processing[13–21]. More impressively, most of POCs with symmetric pore-opening can pack into interconnected channel networks for guest transport independent of the channel's orientation[22]. Recent theoretical calculations have

[1]MOE Key Laboratory of Cluster Science, Beijing Key Laboratory of Photoelectronic/Electrophotonic Conversion Materials, School of Chemistry and Chemical Engineering, Beijing Institute of Technology, Beijing 102488, PR China. [2]State Key Laboratory of Separation Membranes and Membrane Processes, School of Chemical Engineering and Technology, Tiangong University, Tianjin 300387, PR China. [3]Key Laboratory of Nuclear Analysis Techniques, Institute of High Energy Physics, Chinese Academy of Sciences, Beijing 100049, PR China. [4]These authors contributed equally: Si-Hua Liu, Jun-Hao Zhou. ✉e-mail: jiankesun@bit.edu.cn

demonstrated that POCs possess water channel behavior and can provide ultrafast guest permeance and selective separation[23–25]. A promising example was presented by embedding prototypical imine POCs into liposomes that showed aquaporin-like water permeability[22]. However, cell-inspired large-area ultrathin membranes have not been experimentally obtained from these materials. Difficulties may arise from the limited control of the van der Waals packing between cage molecules when phasing out from the dope solution during the membrane formation process[13]; in addition, most of reported POC membranes are based on noncovalent interactions, which unavoidably face the issues of frangibility, defect and instability, especially in the case of large-scale synthesis[26].

Here, we report the successful synthesis of a series of ultrathin networked cage nanofilms with a thickness less than 8 nm using a universal strategy of free-interface-confined self-assembly & crosslinking (FISC) (Fig. 1a). A sharp oil/water (O/W) free interface was introduced to suppress the intrinsic van der Waals packing and direct their 2D self-assembly; the preorganized cage layers were then in-situ crosslinked into continuous ultrathin networked cage nanofilms within the confined 2D space. The nanofilms were able to inherit the nanofluidic channels from the POCs, resulting in exceptional water permeability at a scale of $10^{-5}$ cm$^2$ s$^{-1}$ that surpasses conventional polymeric membranes by 1-2 orders of magnitude. Transferring these networked cage nanofilms onto porous supports affords composite membranes with ultrahigh water permeance up to 360 L m$^{-2}$ h$^{-1}$ bar$^{-1}$ and excellent molecular sieving performance well surpasses that of most current membranes, as well as long-term operational stability. The networked cage composite membrane with adaptive permeance can be achieved by microenvironment modulation (e.g., hydrophilicity and steric hindrance) around the pore windows. Impressively, a light-controlled graded molecular sieving system was established by ionically associating light-responsive anions to the POC membrane, in which the continuous separation of three organic dyes in a single-stage, single-membrane process was exemplified (Supplementary Fig. 1).

## Results

### Preparation and characterization of the ultrathin networked cage nanofilms

A series of mature cycloamine organic cages were synthesized according to the literatures. Their chemical structures were confirmed by $^1$H nuclear magnetic resonance ($^1$H NMR) spectra (Supplementary Figs. 2–5). To realize the FISC process, the first key step is to dissolve the insoluble cycloamine organic cages in water; herein, partial quaternization of amine to enhance the solubility was achieved by adjusting the solution pH with hydrochloric acid (HCl, 0.1 M). As confirmed by electrospray ionization-mass spectrometry (ESI-MS), on average, ~4 of 12 amine groups were protonated in **Cage 1** (Supplementary Fig. 6), while most of the amine groups were maintained. Moreover, the partially protonated cage molecules featuring amphiphilicity favor self-assembly at the sharp O/W interface[27,28] (Supplementary Fig. 7 and Supplementary Movie 1).

As shown in Fig. 1a, the POC nanofilms were synthesized at a free O/W interface with organic cages dissolved in water and trimesoyl chloride (TMC) as the crosslinker dissolved in n-hexane. Free-standing nanofilms formed at an aqueous/organic interface, which qualitatively verified the Schotten-Baumann reaction between the cycloamine cage and TMC (Fig. 1b). A variety of cage molecules (**Cage 1-4**) can be processed into ultrathin nanofilms through this method (Supplementary Figs. 8–10), showing the universality of the FISC strategy. Hereafter, the interfacially assembled and crosslinked cage nanofilm prepared with the prototypical **Cage 1** (Fig. 1a) was selected as the representative material (denoted as the *iac*-cage nanofilm) for detailed investigation. A characteristic band of amide bond at around 1750 cm$^{-1}$ was observed clearly in the Fourier transform infrared (FTIR) spectrum of the *iac*-

cage nanofilm, indicating the formation of amide linkages (Supplementary Fig. 11). High-resolution C1*s* and N1*s* X-ray photoelectron spectroscopy (XPS) further validated the Schotten-Baumann reaction, and revealed that approximately 70 % of the -NH- groups were crosslinked; that is, approximately 8 -NH- groups per cage reacted with acyl chlorides, and the remaining 4 -NH- groups were protonated (Supplementary Figs. 12 and 13). Coupled with the peak ratio of N-C = O and -NH- and the O/N ratio, we calculated that each **Cage 1** molecule in the *iac*-cage nanofilm binds to the other four cages through acyl chloride bridges (Supplementary equations 6 and 7). The nanofilms were sufficiently flexible to be transferred onto various substrates, including nonporous silicon wafers and highly porous supports (e.g., PE, polypropylene, polysulfone and cellulose membranes, Supplementary Figs. 14–16). As illustrated in Fig. 1c, a rectangular *iac*-cage nanofilm with effective area of ~24.0 cm$^2$ was loaded on a porous substrate without any identifiable cracks. Top-surface SEM images of composite membranes demonstrated that all the pores of the PE supports were obstructed by the smooth nanofilms, while the profiles of the substrates were still visible, preliminarily indicating the dense, ultrathin and flexible features (Fig. 1d, e). Then, atomic force microscopy (AFM) was employed to precisely quantify the thickness of the *iac*-cage nanofilms. The height profile reveals that the thickness of the nanofilm is approximately 7.0 nm (Fig. 1f, g). This is at least 10 times thinner than most of state-of-the-art cage films prepared *via* spin coating or interfacial polymerization[26,29,30]. Considering a single **Cage 1** molecule featuring a height of ~2 nm as revealed by molecular simulation (Supplementary Fig. 17), the *iac*-cage nanofilm would be the sum of the thickness of 3 - 4 POC molecules. Furthermore, the thickness of the *iac*-cage nanofilm is tunable by preparation conditions. By increasing the reaction time from 2 to 10 min, the thickness of the nanofilm slowly increased from 6.5 to 8.2 nm (Fig. 1h, Supplementary Fig. 18). Roughness analysis revealed that the nanofilm is super flat with an average roughness below 1.0 nm (Supplementary Table 1). In addition, the nanofilms prepared from other cages showed similar surface roughness and ultrathin feature (Supplementary Figs. 19–21).

Positron annihilation lifetime spectroscopy (PALS) was employed to probe the cavities of the networked cages. Although PALS is a powerful method for analyzing microporous structures in situ[31–33], it is challenging to directly characterize nanofilms with thicknesses less than 10 nm because the signal is too weak to probe. To overcome this limitation, we conducted PALS analysis on bulk-phase crosslinked cages with similar monomer ratios. The results revealed the presence of a micropore with a size around 5.0 Å, suggesting that the intrinsic pore of **Cage 1** was preserved after chemical crosslinking (Supplementary Fig. 22 and Table 2). Molecular simulations were performed to further understand the microstructures of the *iac*-cage nanofilm. As shown in Fig. 1i, the cages are crosslinked into interconnected 3D channel networks that allow water molecules to transport. The simulated pore size distribution in Fig. 1j centered at ~5.0 Å was consistent with the PALs result, further confirming that the networked cage nanofilms inherited the micropores from the parent cages through chemical crosslinking.

### Formation mechanism of the ultrathin networked cage nanofilm

The ultrathin and super smooth features of the cage nanofilms triggered our interest in understanding the underlying membrane formation mechanism. For the traditional interfacial polymerization involving small molecule organic amine and acyl chloride, nucleation and aggregation of polyamide clusters of different molecular weights occur in the organic phase near the oil/water interface, followed by linking into a continuous film via the monomers[34,35]. The thickness and surface roughness will steadily increase under prolonged reaction time, leading to a relatively thick and rough polyamide nanofilm[35,36]. Here, 1,2-diaminocyclohexane, a fragment of **Cage 1**, was used as a monomer to interfacially polymerize with TMC. Under similar

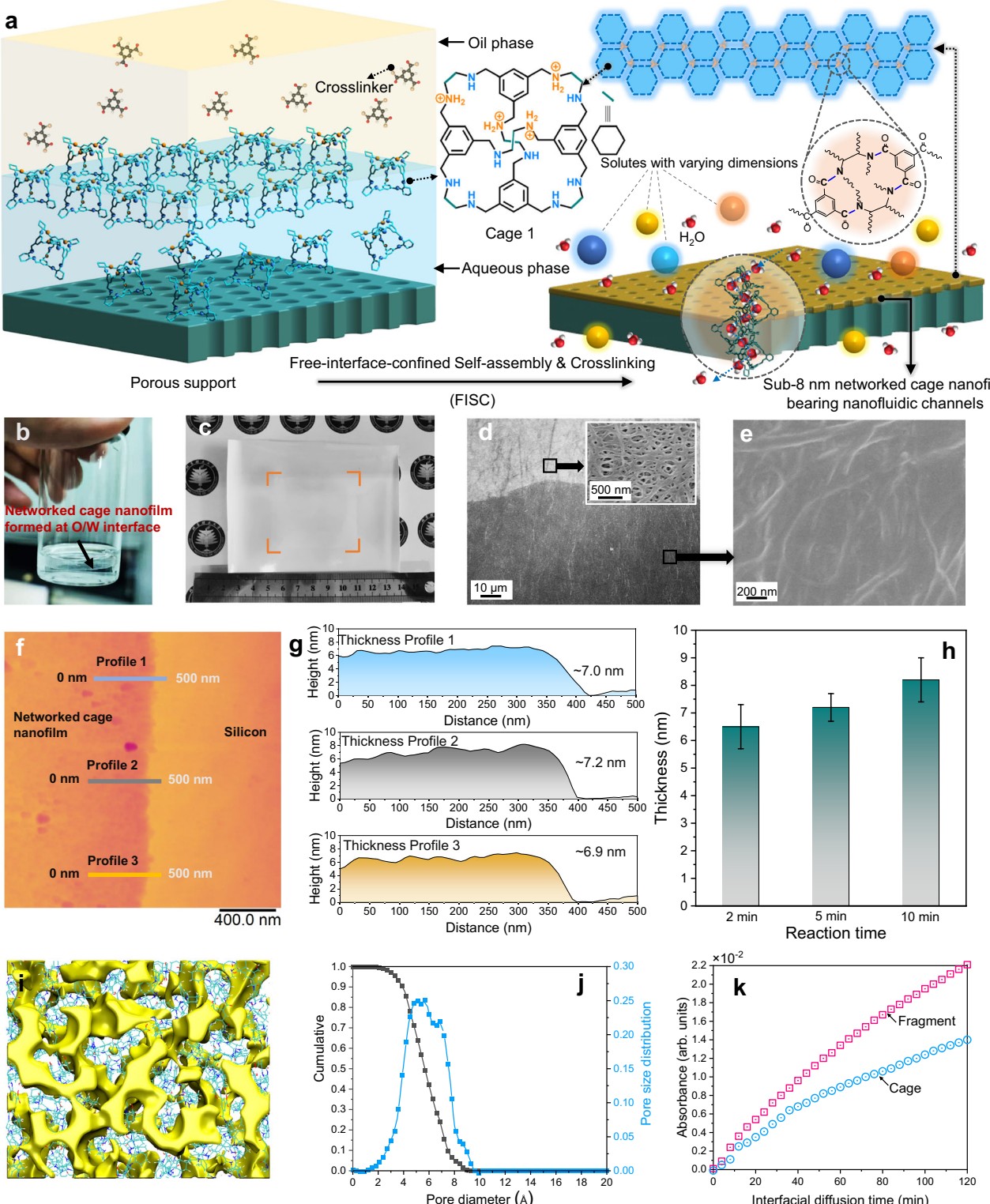

**Fig. 1 | Description of the confined self-assembly and crosslinking process at a sharp free interface and the resulting sub-8 nm networked cage nanofilms.** **a** Crosslinking POCs at a free interface between water and the organic phase by trimesoyl chloride and the schematic diagram of networked cage nanofilm with water channel for rapid molecular separation. **b** Free-standing cage nanofilm formed at the aqueous-organic free interface in a 20 ml vial. **c** Photograph of nanofilm transferred onto polymer substrate. The orange frame indicates the nanofilm boundary. **d** Top-down SEM image of the composite membrane with a substrate of porous polyethylene (PE) ultrafiltration membrane. Inset is a high-magnification SEM image of PE support. **e** High-magnification SEM image of cage nanofilm supported by PE membrane. **f**, **g** AFM height image and corresponding height profile of a section of cage nanofilm on top of a silicon wafer. **h** Thickness increases with the reaction time. Error bars represent the standard deviation calculated from three parallel measurements. **i** 3D view of the cage nanofilm with interconnected pore network. **j** Simulated pore size distribution. **k** UV/vis absorption of the cage and its fragment detected at a position 30 μm away from O/W interface versus interfacial diffusion time.

synthesis conditions, the thickness of the obtained nanofilms increased from ~17 to ~200 nm as the reaction time increased from 2 to 10 min (Supplementary Fig. 23). The same trend was observed for the surface roughness (Supplementary Table 3). The relationship between structure and reaction time aligns with that of conventional polyamide nanofilms (e.g., *m*-phenylenediamine- and piperazine-based poly-amide). However, it sharply contrasts with the behavior of current networked cage nanofilms, which can be attributed to the distinctive chemical structure of the organic cages. When comparing the qua-ternized nitrogen coupled with a hydrophilic counterion (Cl⁻), it becomes evident that the secondary amine group on the cage mole-cule has a higher affinity for the oil phase. This observation aligns with the traditional understanding of interfacial polymerization, where amine groups tend to exhibit some solubility in the oil phase, leading most amine groups to preferentially insert themselves into the oil phase[37,38]. Similar to surfactants, the amphiphilic cage molecules tend to spontaneously assemble at the oil/water interface (with most amine groups inserted in the oil phase, Supplementary Fig. 7c) due to the interfacial energy considerations. Consequently, the uniformly packed organic cage layer will be crosslinked into continuous ultrathin films. Furthermore, the geometric size of the organic cages is 3 times larger than that of 1,2-diaminocyclohexane, leading to a much slower diffu-sion rate from the aqueous phase to the oil phase. We monitored the real-time diffusion kinetics of the organic cage and 1,2-diaminocyclo-hexane across the O/W interface using UV spectroscopy. Evidently, a lower diffusion rate was observed in organic cages than that of 1,2-diaminocyclohexane (Fig. 1k). That is, the confinement of the cage molecules and their assembly at the 2D space of the sharp O/W interface facilitate their subsequent crosslinking into a continuous, ultrathin, and smooth nanofilm.

## Separation performance of the networked cage nanofilm composite membranes

The *iac*-cage nanofilm was loaded onto a porous PE substrate to pre-pare a composite membrane (hereafter denoted as the *iac*-cage membrane), which was then subjected to comprehensive investigation of its molecular nanofiltration performance using a range of dye aqu-eous solutions. To confirm that dye adsorption did not affect the membrane selectivity, dye adsorption tests using Valia-Chien diffusion cell were performed. No noticeable variation in UV-vis absorption was observed in the feed chamber, and the permeate chamber remained colorless even after 72 h, indicating that the membrane did not absorb the dyes (Supplementary Fig. 24). All dye separation experiments were performed using a high concentration feed of 100 mg mL⁻¹, and all the results were recorded after at least 30 min of steady filtration. For the *iac*-cage membranes prepared with cage concentrations of 1.0, 4.0 and 8.0 mM, the water permeance increased with decreasing reaction time (Fig. 2a), which is consistent with the observed variation in nanofilm thickness, as demonstrated by AFM analysis. Significantly, the *iac*-cage membrane prepared with a cage concentration of 1.0 mM and a reac-tion time of 2 min exhibits an exceptionally high water permeance of 360 L m⁻² h⁻¹ bar⁻¹ (the corresponding rejection for Congo red was 99.0%), which is 1–2 orders of magnitude higher than that of com-mercial nanofiltration membranes. Figure 2b shows the rejection behavior of the membrane towards a range of dye molecules with different geometric sizes (Supplementary Fig. 25). Interestingly, the *iac*-cage membranes displayed strict size-dependent selectivity. Almost all the dye molecules with dimensions larger than 5.2 Å can be rejected by the membrane, while the rejection to the molecule with a dimension below 4.8 Å sharply decreased to as low as 10.0% (for the 1 mM-2 min membrane, Fig. 2b). The sharp size-sieving behavior should be attributed to the sieving effect of cages. In addition, the electrostatic interaction slightly influences the rejection behavior. For the dyes with similar sizes but opposite charges, the membrane dis-played a rejection order of Congo red (CR) > Alcian blue (AB) and

Methyl orange (MO) > Methylene blue (MB), i.e., the rejection of negatively charged dye was relatively higher than that of positively charged one (Supplementary Fig. 26). Such phenomenon is consistent with the Donnan effect[39,40], since the membrane top surface is nega-tively charged (−3.0 mV) at a neutral pH (Supplementary Fig. 27). As shown in Fig. 2c, the nanofiltration performance of the *iac*-cage membranes in the aqueous environment surpasses that of most commercial and documentary membranes to date.

Moreover, such outstanding separation performance was largely maintained with increasing operation pressure up to 5 bar and over 120 h of the cross-flow filtration test, as well as upon immersion in strong acid aqueous solutions, and polar/nonpolar organic solvents (Supplementary Figs. 28, 29), highlighting their high structural stability.

To emphasize the unique role of the cage cavity in water transport, a control membrane without intrinsic pores was prepared using a fragment of **Cage 1** as a monomer under similar conditions. The control membrane displayed a water permeance of 5.5 L m⁻² h⁻¹ bar⁻¹, which is 65-fold lower than that of the *iac*-cage membrane, although it should be noted that it was around 10 times thicker. Furthermore, we calculated the intrinsic water transport character-istics, i.e., water permeability, $P_w$, of the *iac*-cage membrane, and compared them with those of the control membrane and state-of-the-art membrane materials[41]. As shown in Fig. 2f, the $P_w$ value of the *iac*-cage membrane is up to 10⁻⁵ cm s⁻¹, which is one to two orders of magnitude higher than that of the control membrane and traditional polymeric membranes, and is comparable to that of nanofluidic membranes (e.g., MOFs, COFs and GO membranes)[9]. Very recently, He et al. reported a crystalline imine cage membrane with a minimal thickness of ~80 nm and a moderate water permeance of 43.0 L m⁻² h⁻¹ bar⁻¹ [42]; the calculated $P_w$ value of this membrane is of a similar magnitude to that of our *iac*-cage membrane. These results indi-cated the preservation of nanofluidic water channels of cages within the *iac*-cage membrane. This also highlights a significant advantage of the FISC method, through which the amine cages can be inter-facially crosslinked into sub-8 nm cage nanofilms, simultaneously largely preserving the nanofluidic water channels and thus leading to outstanding molecular separation performance (Fig. 2c). It is also worth mentioning that scale-up preparation of the *iac*-cage mem-brane might be achieved by engineering the FISC method into a roll-to-roll technique[43,44].

## Tuning the nanofluidic water flow through microenvironment modulation

**Regulation of pore window hydrophilicity.** Controllable regulation of the water transport behavior is a significant feature of cell membrane channels[3]. The rich chemistry of the organic cage offers promising opportunities to regulate the nanofluidic water flow via modulating the microenvironment surrounding the cage window, e.g., previous studies have shown that the hydrophilicity/hydrophobicity of cationic cages can be tuned through anion metathesis[45,46], potentially enabling the modulation of channel affinity toward water molecules. Fortunately, there are four cationic N atoms per cage molecule in the *iac*-cage membrane with hydrophilic Cl⁻ as the counteranion according to XPS analysis, which can be exchanged with a hydrophobic one, i.e., bis(tri-fluoromethane sulfonyl)imide (TFSI⁻) (Fig. 3a). The energy-dispersive X-ray spectroscopy (EDS) mapping results demonstrated the even dis-tribution of Cl on the membrane surface (Supplementary Fig. 30 and Table 4). The Cl/N ratio was around 0.33, further supporting the pre-sence of ~4 cationic N atoms per cage molecule. After exchange with TFSI⁻, the Cl element is hardly detectable, while the feature elements of F and S appear and are homogeneously distributed on the membrane surface. Meanwhile, the water contact angle of the membrane increased from 51.3° ± 0.8° to 62.3° ± 1.5° (Supplementary Fig. 31).

A more hydrophobic separation membrane typically has difficulty forming a water layer on its surface, leading to a decrease in water

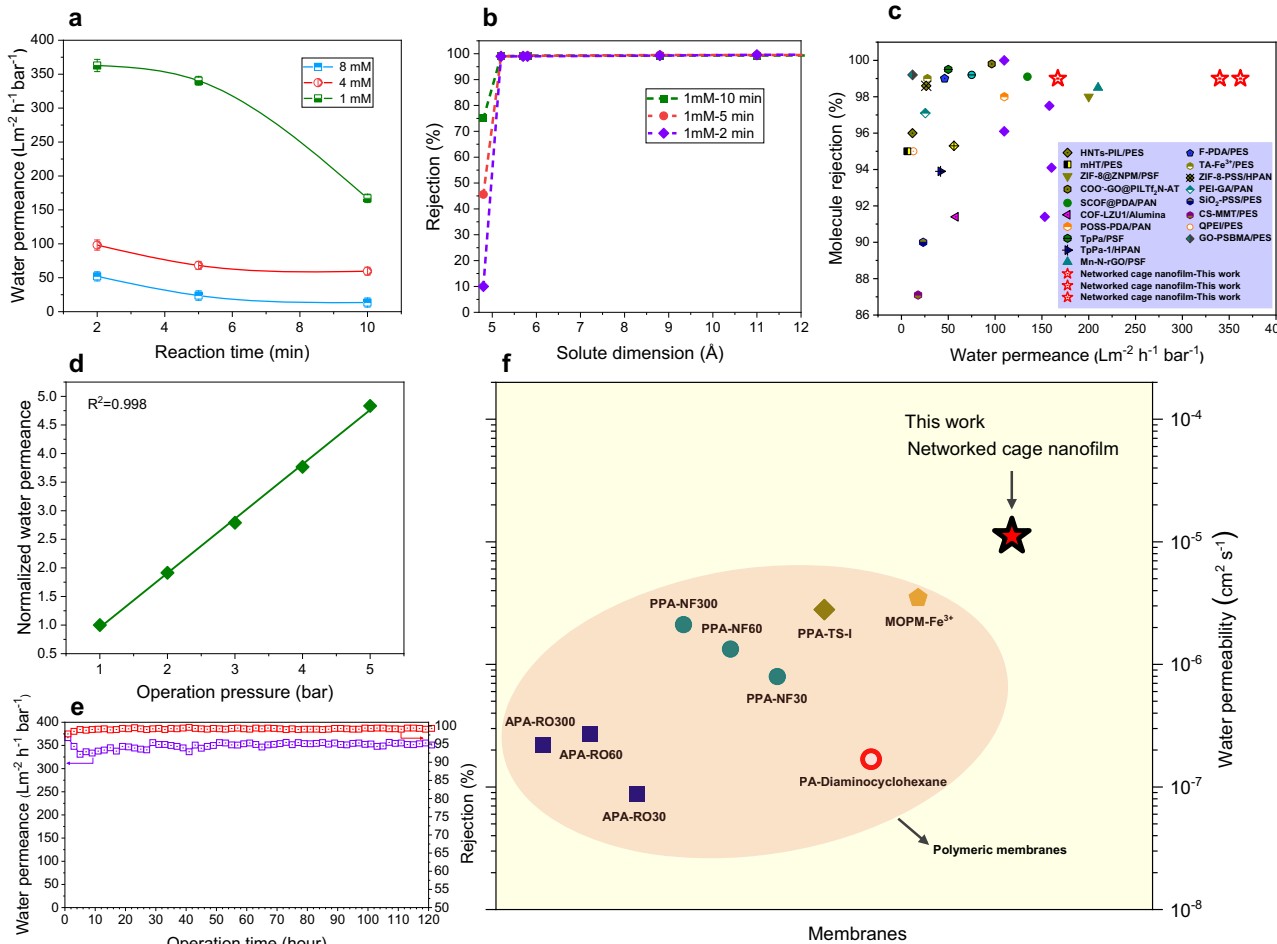

**Fig. 2 | Separation performance. a** Water permeance of networked cage composite membranes varied with synthesis conditions. Error bars represent the standard deviation calculated from three parallel membrane samples. **b** Rejection of different cage composite membranes versus the dimensions of the dye molecules. **c** Aqueous nanofiltration performance comparison of cage composite membranes with commercial and literature reported membranes. The molecule rejection data refer to selectivity of Congo red. **d** Water permeance with increasing operation pressure. **e** Continuous filtration at an operation pressure of 1 bar. **f** Water permeability comparison of cage composite membranes with traditional polymeric membranes.

permeance[47]. Counterintuitively, as depicted in Fig. 3b, the cage membrane paired with hydrophobic counterions TFSI⁻ exhibited the highest water permeance and simultaneously maintained high CR rejection (99.0%). The calculated $P_w$ value is ~1.2 times higher compared to the membranes carrying Cl⁻. This result could be attributed to the weak interactions between water molecules and hydrophobic groups around the cage window, which reduces the time required for water molecules to traverse the cage, thereby improving the membrane water permeance[5,48,49]. To gain further insights, molecular dynamics (MD) simulations were performed using an ~8.0 nm-thick cage nanofilm with different anions (Fig. 3g, h, Supplementary Fig. 32, Movies 2 and 3). The steady water permeability was calculated to be $3.52 \times 10^{-5}$ cm² s⁻¹, a value comparable to the experimental result of $1.11 \times 10^{-5}$ cm² s⁻¹. The cage networks in Fig. 3g show 1D chains of water molecules, which is a typical feature reported in biological water channels[1,50]. The diffusive ability of water molecules was analyzed by tracking the number of water molecules transferred across these networked cage membranes in Fig. 3h. The results clearly reveal that water molecules diffuse faster in the TFSI⁻ exchanged cage membrane, consistent with the observed separation performance in Fig. 3b.

**Modulation of window opening size with light-responsive counteranions.** The facile counteranion exchange strategy inspired us to introduce molecular conformation-responsive counteranions to the cage window, which mimics the gating effect of the cell membrane, where the cationic N serves as the "gating hinge" and the environmentally adaptive counteranions acts as the "gates". As a proof-of-concept, azobenzoate (azo), the typical light-responsive molecule[51,52], was selected as the smart counteranion, which endows the cage membrane with a photo-responsive gating effect (Fig. 3a). The experiment started with counteranion exchange. i.e., replacing Cl⁻ with an azo anion. The *iac*-cage membrane carrying the azo anion is denoted as the *iac*-cage-azo membrane hereafter. Successful anion exchange was confirmed by EDS mapping (Supplementary Fig. 34). The photo-induced conformational change (i.e., photoisomerization) of azo in the *iac*-cage-azo membrane was analyzed through solid UV spectroscopy. As shown in Fig. 3c-e, after UV irradiation, the absorbance (~322 nm) assigned to the bonding-antibonding orbital [π-π*] transition of the *trans*-azo moiety decreases remarkably in intensity, while a slight increase in the band at ~430 nm is observed (assigned to the n-π*transition of the *cis*-azo moiety). The photo-stationary state of the *iac*-cage-azo membrane can be achieved after 10 min of irradiation, and the proportion of *trans*-azo in the *iac*-cage-azo membrane varied from 100% to ~70%, as determined by UV spectroscopy. Such a process is reversible upon irradiation with visible light, as evidenced by the 6 cycles of alternating UV and Vis irradiation. By comparison, the pristine *iac*-cage membrane carrying Cl⁻ anions is inert to photoirradiation

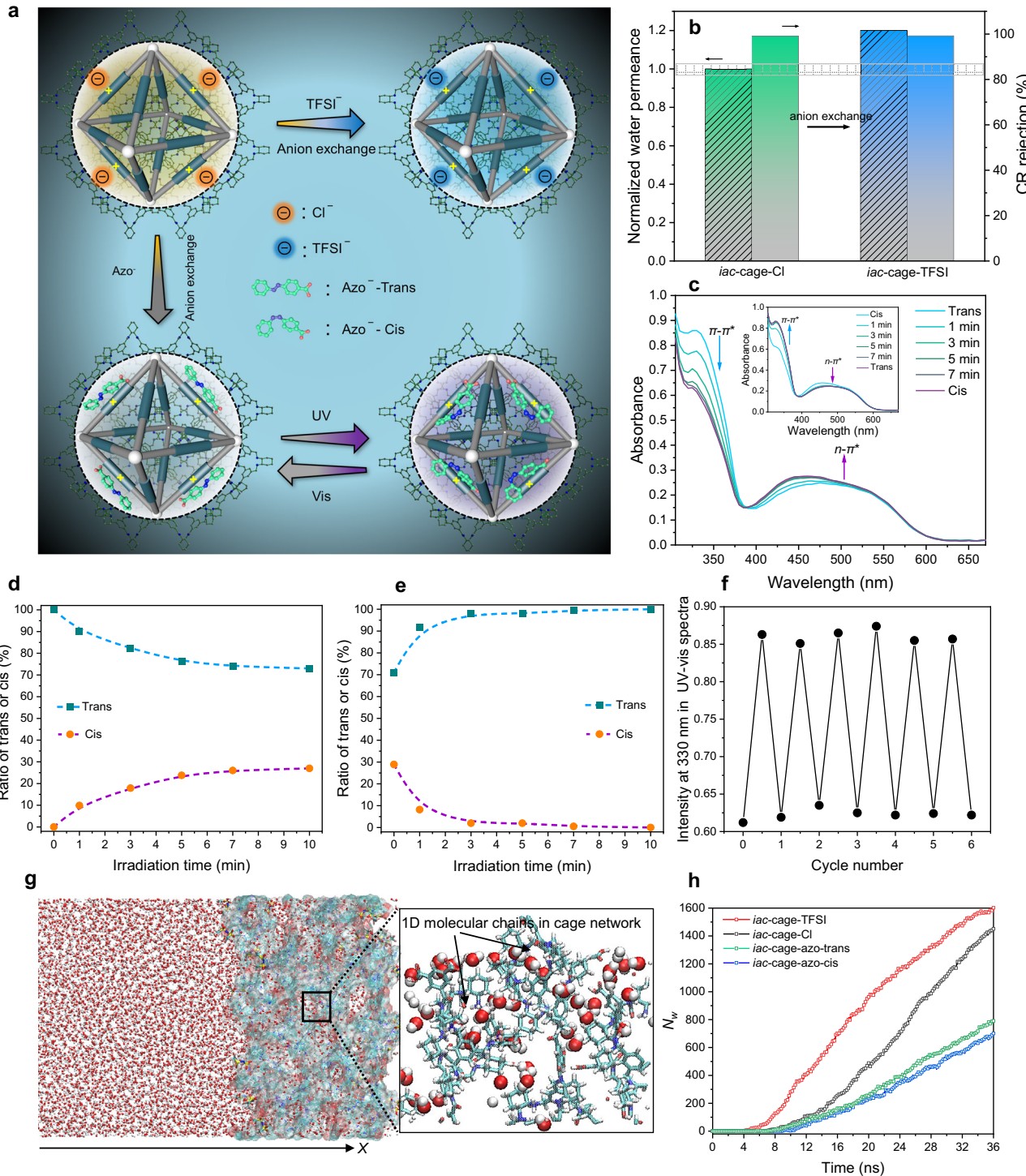

**Fig. 3 | Microenvironment modulation for regulating water flow and simulation. a** Schematic illustration showing the microenvironment modulation of the pore windows via anion exchange. **b** Separation performance enhancement after TFSI⁻ exchange. **c** Time-dependent UV/Vis spectra of the *iac*-cage-azo membrane at 298 K, using UV and Vis (inset) irradiation. **d**, **e** Ratio of trans/cis state with UV and Vis light irradiation time, respectively. **f** Changes of the absorption band of *iac*-cage-azo membrane at 330 nm upon alternating irradiation by UV and visible light. **g** Simulation snapshot of water molecules transporting through the *iac*-cage-TFSI membrane. The enlarged snapshot shows the water chains inside the cage network. **h** Numbers of transferred water molecules $N_w$ through cage membranes with various counterions.

under the same conditions, and no obvious UV-Vis absorbance variation can be detected (Supplementary Fig. 35).

The separation performance evolution of the *iac*-cage-azo membrane under alternating UV and Vis irradiation was investigated. After 10 min of UV irradiation, a 20% decrease in water permeance was observed, whereas an almost complete rejection of CR was

maintained; upon subsequent 10 min of Vis irradiation, the water permeance recovered and maintained a CR rejection of 99.0%. MD simulations also demonstrated that water molecules diffuse slower within the *iac*-azo-cis-nanofilm compared to the *iac*-azo-trans-nanofilm (Fig. 3h, Supplementary Fig. 33, Movie 4 and 5). We attribute this reversible water permeance to the variation in steric hindrance with

photoisomerization of azo. We further investigated how light control the window opening sizes by measuring the rejection of a range of dyes. As shown in Fig. 4b, the *iac*-cage-azo membrane upon Vis irradiation shows a cut-off rejection (≥ 90%) around 5.1 Å; when the membrane is treated with UV light, the cut-off rejection slightly decreases to 4.5 Å. The shrinking of cut-off rejection experimentally evidenced the light-controlled window opening sizes. MD simulation results also confirms that projected pore aperture of the **cage 1** equipped with *cis*-azo is smaller than that with *trans*-azo (Supplementary Fig. 36). More interestingly, as depicted in Fig. 4c, rejections towards dyes with relatively smaller dimension (e.g., methyl orange and 4-nitrophenol) are switchable in each cycle of UV/Vis irradiation, indicative of adaptive separation performance.

### Photo-responsive Graded molecular separation

Adaptive molecular sieving membrane provide an opportunity for separating ternary systems or more complex mixtures using a single membrane[42,51]. The *iac*-cage-azo membranes displaying switchable water permeance and sieving performance hold promise for graded molecular sieving. As a proof-of-concept, the *iac*-cage-azo membrane was employed to separate a mixture of three organic dyes with gradient dimensions (i.e., 4-nitrophenol, 2.5 Å, methyl orange, 4.8 Å, and Congo red, 5.2 Å). Upon UV-irradiation, only 4-nitrophenol was detected in the permeate (Fig. 4d), and Congo red and methyl orange were nearly completely rejected. After Vis irradiation, the methyl orange became permeable while the Congo continued to be rejected, as confirmed by the pure phase of methyl orange in the permeate (Fig. 4e). After flushing the residual methyl orange from the feed with excess deionized water, the pure phase of Congo red could be collected (Fig. 4f). Upon alternating UV and Vis irradiation, a single *iac*-cage-azo membrane achieved the separation of a ternary mixture. The *iac*-cage-azo membrane is recyclable and robust, as evidenced by at least 5 cycles of operation, consistent with the results obtained from solid UV spectroscopy.

## Discussion

In summary, by confining the organic cage molecules at the oil/water interface, a range of cage molecules were self-assembled and then chemically crosslinked into flexible nanofilms with thicknesses down to 8 nm. The resulting composite membrane exhibited nanofluidic channels, enabling exceptional water permeance that surpassed commercial nanofiltration membranes by 1–2 orders of magnitude. Furthermore, the organic cage membrane exhibited a wide range of chemical functionalities, offering the possibility of facile micro-environmental modulation of the nanofluidic water channels. The membrane even displayed a photo-responsive gating effect, enabling photo-controlled graded separation. Given the versatility of POCs, these networked cage ultrathin nanofilms hold great potential for a wide range of future applications in catalysis, sorption, and sensing, particularly with task-specific functionalization.

## Methods

### Chemicals and materials

(*R,R*)−1,2-diaminocyclohexane (99.0%), 1,2-ethylenediamine (99.0%), 1,3,5-triformylbenzene (99.0%), 2-hydroxy−1,3,5-triformylbenzene (99.0%), sodium borohydride (NaBH$_4$) (98.0%), trifluoroacetic acid (99.0%), hydrochloric acid (36.0–38.0wt.% in water), trimesoyl chloride (99.0%), lithium bis(trifluoromethane sulfonyl)imide (LiTFSI) (99.5%), sodium azobenzoate (99.0%), 4-nitrophenol (indicator), methyl orange (indicator), Congo red (indicator), methyl red (indicator), methylene blue (indicator), and all solvents were obtained from commercial sources and directly used without further purification. Porous polyethylene (PE) ultrafiltration membranes were supplied by OriginWater, P. R. China. The polysulfone and cellulose membranes were purchased from Tianjin JinTeng, P. R. China.

### Synthesis of Cage 1

Typically, dichloromethane (10 mL) was added slowly to 1,3,5-triformylbenzene (500 mg). Trifluoroacetic acid (10 μL) was added to this solution as a catalyst. Then, a dichloromethane solution (10 mL) of (*R,R*)−1,2-diaminocyclohexane (500 mg) was added. The mixture was capped and left to stand for one week, during which, crystals formed on the sides of the vessel. The crystalline product was collected by centrifugation and washed with a dichloromethane/methanol solution (v/v = 5/95) for 3 times (50 mL × 3), and further dried at 100 °C for 24 h. After that, the imine cage (463 mg) was dissolved in a dichloromethane/methanol mixture (v/v = 1/1, 25 mL) under vigorous stirring. When the solution became clear, NaBH$_4$ (500 mg) was directly added into the solution and kept stirring for 15 h. 1 mL of water (1 mL) was then injected and the mixture was kept stirring for another 9 h. Finally, the solution was removed under vacuum. The residual was washed with a large amount of water for 3 times (100 mL × 3) until it became neutral. All the reaction processes were conducted at room temperature. The resulting sample was then vacuum dried at 80 °C for 24 h and stored in a light-resistant manner to afford **Cage 1** as white powder.

### Synthesis of Cage 2

2-hydroxy-1,3,5-triformylbenzene (100 mg) and (*R,R*)−1,2-diaminocyclohexane (100 mg) were dissolved in DMF solvent (25 mL) and heated at 90 °C for 3 days in a Teflon-lined stainless steel autoclave (100 mL). After slowly cooling to room temperature, the yellow crystalline product was collected and washed with acetone for 3 times (50 mL × 3), then vacuum dried at 80 °C for 24 h to yield imine cage as a fine orange powder. After that, the imine cage (50 mg) was dissolved in dichloromethane/methanol solution (v/v = 1/1, 10 mL) under vigorous stirring. When the solution became clear, NaBH$_4$ (100 mg) was directly added into the solution and reacted for 15 h. 1 mL of water (1 mL) was then injected and the mixture was kept stirring for another 9 h. Finally, the solution was removed under vacuum. The residual was washed with a large amount of water for 3 times (20 mL × 3) until it became neutral. The resulting sample was then vacuum dried at 80 °C for 24 h and stored in a light-resistant manner to afford **Cage 2** as pale-yellow powder.

### Synthesis of Cage 3

Ethyl acetate (35 mL) was added to 1,3,5-triformylbenzene (50 mg) in a beaker at room temperature. After 5 min, a solution of 1,2-ethylenediamine (28 mg) in ethyl acetate (5 mL) was added. The mixture was capped and left to stand for 60 h without stirring. Pale-white needle-like crystals were observed after around 60 h. The crystals were collected carefully from the sides of the flask and washed with ethyl acetate for 3 times (20 mL × 3), and then vacuum dried at 80 °C for 12 h to yield imine cage as a white powder. The reduction process is similar to that of **Cage 2**.

### Synthesis of Cage 4

Tetraaldehyde 2 (150 mg) and 2 equiv KOH were dissolved in a mixture of water and ethanol (100/100 ml, v/v) and refluxed for 1 h. A 50 mL of ethanol solution of (*R,R*)−1,2-diaminocyclohexane (137 mg) was added into the mixture, and refluxed for additional 24 h. The resulting clear solution was filtered and the filtrate was allowed to slow evaporation at approximate 30 °C to afford cage crystals. The reduction process is similar to that of **Cage 2**.

The molecular structures of all the cage compounds were confirmed through $^1$H NMR analysis using CDCl3 as the deuterated solvent (Supplementary Figs. 2–5).

### Fabrication of free-standing networked cage nanofilm

The free-standing cage nanofilms were prepared through a free-interface-confined self-assembly and crosslinking strategy (FISC). First, aqueous solutions of imine cage were prepared through a partial

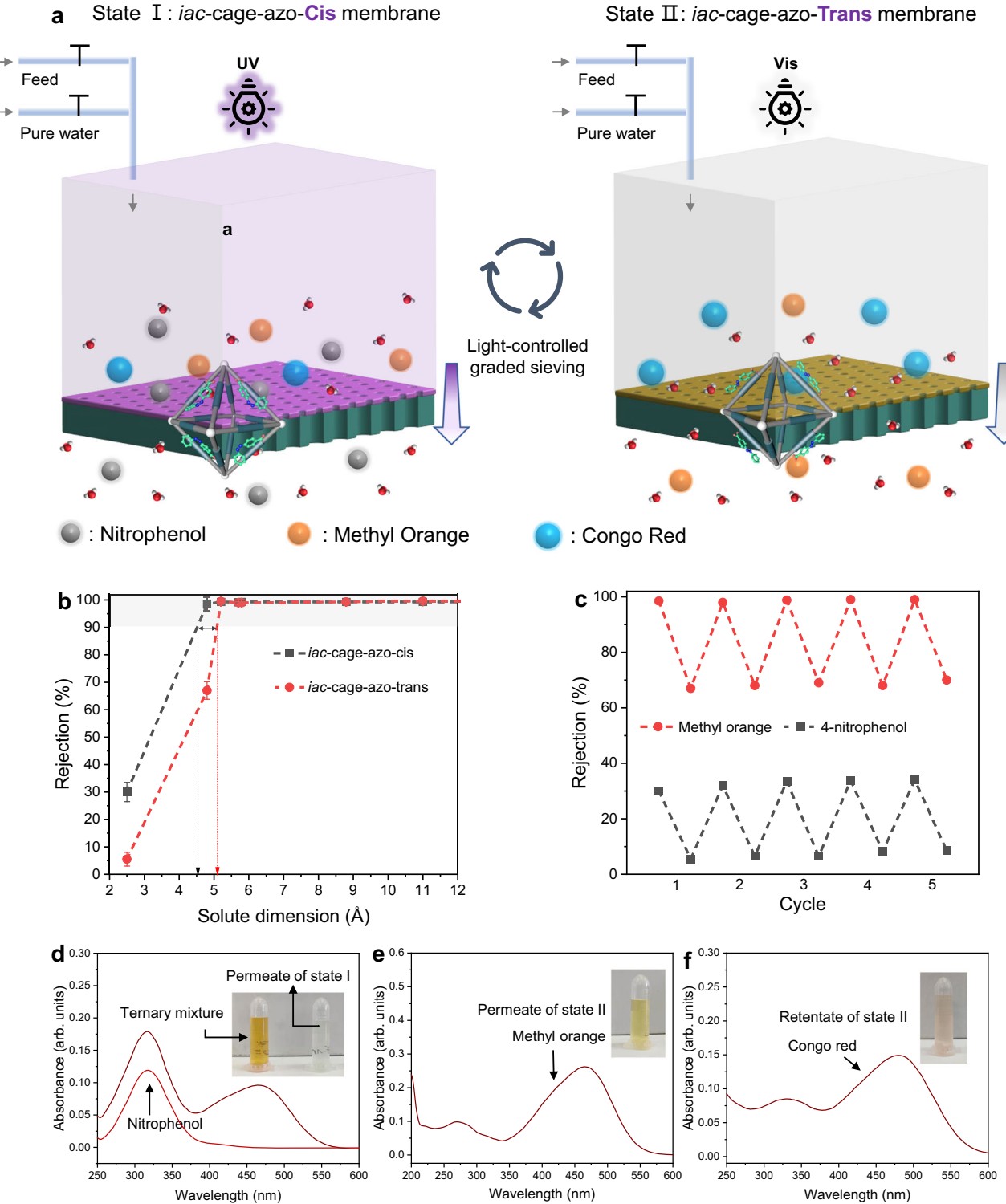

**Fig. 4 | Photo controlled graded molecular separation. a** Schematic demonstrating the light-controlled graded molecular sieving. **b** Molecule rejections of the *iac*-cage-azo membrane with increasing solute dimension after UV and Vis irradiation. The error bars show the standard deviation of three parallel membrane samples. **c** Reversible dye rejection of methyl orange and 4-nitrophenol of the *iac*-cage-azo membrane observed upon alternating UV and Vis irradiation. **d** UV-Vis absorption spectra of the ternary mixture in water and the permeate from the membrane of state I. **e** UV-Vis absorption spectra of the permeate from the membrane of state II, showing the existence of methyl orange. **f** UV-Vis absorption spectra of the retentate of state II, showing the pure phase of Congo red. Inserts are the digital pictures related to permeate and retentate. The standard UV-Vis absorption spectra of 4-nitrophenol, methyl orange and Congo red are presented in Supplementary Fig. 25.

quaternization method. Imine cages were added to water (2 ml) and sonicated to form a turbid dispersion. Subsequently, HCl (0.1 M) was added dropwise until the pH reached approximately 8.0, resulting in a nearly transparent solution. The solution was then filtered using a syringe filter (pore size: 0.22 μm) and transferred to a glass dish. To initiate the FISC process, 1 ml of TMC solution in n-hexane was carefully added to the water surface. The reaction proceeded for a specific duration, after which the resulting cage nanofilms were transferred to a silicon wafer disc and rinsed with a solvent. The prepared samples were further heat-treated at 60 °C for 5 min for further characterization. The concentrations of the cage aqueous solution ranged from 1 mM to 8 mM, and the reaction time varied from 2 min to 10 min. The concentration of the TMC organic solution remained constant at 6 mM.

### Fabrication of networked cage nanofilm composite membrane
For the fabrication of composite membrane, the clear cage aqueous solution was added onto a pre-clamped porous substrate positioned between a vacuum filter head and a vacuum filter bowl. Once the reaction was complete, the cage aqueous solution was filtered to facilitate the adhesion of the formed cage nanofilm to the porous substrate. Subsequently, the oil phase solution was poured off and the resulting composite membrane was heat-treated at 60 °C for 5 min. Finally, the composite membrane was rinsed with n-hexane to remove residual TMC.

### Counterion exchange for the networked cage nanofilms
A simple anion exchange strategy was employed to manipulate the channel microenvironments including hydrophilicity and steric hindrance. All the counterion exchange experiments were started with an *iac*-cage-Cl-membrane. For the *iac*-cage-TFSI-membrane, 30 ml aqueous solution of TFSILi (10 mg ml⁻¹) filtrated through the *iac*-cage-Cl-membrane and rinsed with deionized water (3*30 ml). For the *iac*-cage-azo-membrane, 30 ml aqueous solution of sodium azobenzoate (10 mg ml⁻¹) was used.

For more information on separation performance experiments, property characterizations, and computational simulations, please refer to Supplementary Information.

## Data availability
The main data generated in this study are provided in the article and Supplementary Information. Additional data are available from the corresponding author on request.

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

## Acknowledgements

The financial support for this work was provided by National Natural Science Foundation of China (22071008, 22208018, 52003029), the High-level Overseas Talents Program of China, the Excellent Young Scholars Research Fund from the Beijing Institute of Technology, and the Central University Basic Research Fund of China (2021CX01024). The authors acknowledge the staff at the Analysis & Testing Center, Beijing Institute of Technology for their helpful technical supports and discussions.

## Author contributions

J.-K.S and S.-H.L. conceived and designed the experiments. S.-H.L. performed the synthesis of cage and membrane materials, characterizations and separation performance tests. X.C. and P.Z. performed the PALs analysis. J.-H.Z. performed all the simulations. C.W. provided constructive suggestions. All authors were involved in the analysis and discussion of the results. J.-K.S., S.-H.L. and J.-H.Z. wrote the manuscript and Supplementary Information.

## Competing interests

The authors declare no competing interests.
