## [Peer Review File · Nature Communications]

REVIEWER COMMENTS

Reviewer #1 (Remarks to the Author):

Please refer to the attached reviewer comments.

Reviewer #2 (Remarks to the Author):

In this study, the authors reported an interfacial polymerization technique to fabricate ultrathin membranes from porous cage materials. The researchers showcased the membranes' remarkable water flux. They further demonstrated that by ion exchanging photosensitive ligands, the membrane flux can be regulated through light modulation. I have the following inquiries.

(1) The primary concern in this study lies in the underestimated size of the dye molecules. The majority of the dye molecules reported in this work should be approximately 1 nm instead of 0.5 nm. This discrepancy raises significant doubts regarding the actual contribution of the porous cage materials to the membrane's functionality, considering that the window size of the cages is merely 0.5 nm. To address this concern, the authors must provide solid evidence that confirms the transport occurs via the porous cage structure rather than through membrane defects.

(2) From Figure 2a, I observed a substantial decrease in water permeance as the reaction time increases or when higher concentrated solutions are utilized. However, in Figure 1h, the membrane thickness does not appear to increase to the same extent. This trend further deepens my concerns regarding the true impact of the porous cage materials on the membrane's functionality. Therefore, it is essential for the authors to offer a more comprehensive study on the membrane structure.

(3) In the reported pore size range, i.e. ~ 0.5 nm, the membrane should be able to reject monovalent ions such as Na^+ , or at least reject divalent ions such as Mg^{2+} . The authors should demonstrate the membrane performance in these systems instead of dye molecules.

(4) Was the reported water permeance measured using pure water or a dye solution? Similarly, regarding the long-term permeation test, is it also conducted with pure water? If the measurements were performed solely with pure water, the long-term test may not offer valuable information on fouling and consequently may have limited usefulness.

(5) Regarding the measurement of dye rejection, although the authors acknowledged the potential adsorption effect by allowing a half-hour waiting period prior to measurement, this duration may be insufficient given the high surface area of the porous cage materials and the relatively low dye concentration. To address this concern adequately, it is necessary for the authors to conduct an adsorption experiment to quantify the adsorption capacity of each dye.

(6) The authors utilized a chemistry similar to polyamide membranes in crafting the porous cage membrane, wherein the porous cage molecules served as imine moieties. While the authors claim that the conventional process cannot yield ultrathin membranes, this assertion is not entirely accurate. It would be advisable for the authors to refer to reference (Science 2015, 348, 1347–1351) to appreciate that ultrathin membranes can indeed be achieved through meticulous process control. It is worth noting that due to the disparity in membrane thickness, the reported performance improvement may be somewhat exaggerated. For instance, although the authors mentioned a 65-fold reduction in permeance for membranes produced through the conventional process, but these membranes are also 10 times thicker. Moreover, an important point omitted by the authors is that membranes produced through the conventional process are much tighter, making them suitable for reverse osmosis (RO) or nanofiltration (NF) applications.

(7) Standalone membranes are generally not preferred in practical applications due to their limited mechanical strength. Therefore, I suggest that the authors consider employing a similar process as that used for reverse osmosis (RO) or nanofiltration membranes to create a thin film on a porous support. If such an approach is pursued and the membrane performance exhibits a comparable improvement, it would greatly enhance my confidence in supporting this work.

(8) The authors stated that the cage molecules are aligned at the interface. However, it is crucial for them to present comprehensive characterization data to substantiate this claim.

(9) The authors solely focused on investigating water permeation in their study. However, it would be highly beneficial to extend their investigation to include other commonly used solvents such as methanol, ethanol, NMP, hexane, and so on. By studying the permeation behavior of these additional solvents, a more comprehensive understanding of the membrane's performance can be gained.

Reviewer #3 (Remarks to the Author):

This is an interesting manuscript based on the use of cages for the preparation of thin film composite membranes.

My first concern is why the authors do not mention previous manuscripts using porous organic cages for the preparation of thin film composite membranes using interfacial polymerization.

Examples are Li et al. Nature Comm. 14, 3112 (2023) with a quite similar approach or He et al. Nature Mater. 21, 463 (2022).

As for light-sensitive membranes with azo segments, the authors probably know the paper of Liu et al. Science Adv. 6, eabb3188, 2020.

I do not understand Fig 2f. What is the x-axis?

Fig. 2c: molecular rejection of which molecule size?

Fig. 2b: what is the cut off in g mol^{-1} ?

More information on the set-up used for the UV permeance experiments is needed.

This manuscript reported a series of ultrathin networked porous organic cage (POC) nanofilms with a thickness of sub-8 nm by the free-interface confined self-assembly and crosslinking method. The obtained porous substrate supported POC membranes of intrinsic nanofluidic channels show an ultrahigh water permeance up to $360 \text{ L m}^{-2} \text{ h}^{-1} \text{ bar}^{-1}$ and excellent molecular sieving performance. In addition, the microenvironment (e.g., hydrophilicity and steric hindrance) of POC pore windows can be modulated to achieve a light-controlled graded molecular sieving. This work is novel and the experimental data and theoretical analysis are sufficient. Some specific comments are listed and should be addressed.

(1) The PALS results of Cage 1 revealed the presence of a micropore with a size around 5.0 \AA . What does the measured size stand for, the window size or the cavity size of Cage 1?

(2) “The simulated pore size distribution in Figure 1j centered at $\sim 5.0 \text{ \AA}$ was consistent with the PALS result”. How is the pore size distribution of Cage 1 simulated?

(3) “Driven by the interfacial energy, a layer of amphiphilic organic cages self-assembles at the oil/water interface, with most amine groups inserted in the oil phase prior to reacting with the TMC crosslinkers”. Why do most amine groups insert in the oil phase? How is the whole process driven by the interfacial energy? A detailed explanation should be added.

(4) The membrane is positively charged, since ~ 4 of 12 amine groups are protonated. However, as shown in the surface zeta potential of the cage composite membrane, the membrane is negatively charged (-3.0 mV) at a neutral pH. A reason should be given.

(5) The hydrophobic groups are around the cage window, so how can they reduce the time required for water molecules to traverse the cage and improve the membrane water permeance?

(6) There is no discussion about the crystallinity of the prepared POC membranes throughout the manuscript, why?

Response to Reviewer

Reviewer 1#

This manuscript reported a series of ultrathin networked porous organic cage (POC) nanofilms with a thickness of sub-8 nm by the free-interface confined self-assembly and crosslinking method. The obtained porous substrate supported POC membranes of intrinsic nanofluidic channels show an ultrahigh water permeance up to $360 \text{ L m}^{-2} \text{ h}^{-1} \text{ bar}^{-1}$ and excellent molecular sieving performance. In addition, the microenvironment (e.g., hydrophilicity and steric hindrance) of POC pore windows can be modulated to achieve a light-controlled graded molecular sieving. This work is novel and the experimental data and theoretical analysis are sufficient. Some specific comments are listed and should be addressed.

1. The PALS results of Cage 1 revealed the presence of a micropore with a size around 5.0 \AA . What does the measured size stand for, the window size or the cavity size of Cage 1?

Reply: Thank you for your question regarding the interpretation of the PALS results. Typically, PALS analysis provides insights into the average size of free volume within a material (*Chem. Sci.* **2022**, 13, 5042; *Angew. Chem. Int. Ed.* **2019**, 58, 2638). In the context of our study, this free volume measurement can be regarded as representative of the cavity within the cage; these results suggest that the intrinsic pore of the cage was preserved after the chemical crosslinking process.

2. “The simulated pore size distribution in Figure 1j centered at $\sim 5.0 \text{ \AA}$ was consistent with the PALS result”. How is the pore size distribution of Cage 1 simulated?

Reply: Thank you for your inquiry. The pore size distribution of Cage 1's network was analyzed using Zeo++, based on a cage network model constructed from the chemical composition obtained through XPS analysis. The detailed simulation information was provided in the Supplementary Information (Section 2.5, page 10, highlighted with a yellow background).

3. “Driven by the interfacial energy, a layer of amphiphilic organic cages self assembles at the oil/water interface, with most amine groups inserted in the oil phase prior to reacting with the TMC crosslinkers”. Why do most amine groups insert in the oil phase? How is the whole process driven by the interfacial energy? A detailed explanation should be added.

Reply: We greatly appreciate your valuable comments. Detailed explanation was added in the revised manuscript (In the main text, paragraph 2, page 7, highlighted with a yellow background):

“When comparing the quaternized nitrogen coupled with a hydrophilic counterion (Cl^-), it becomes evident that the secondary amine group on the cage molecule has a higher affinity for the oil phase. This observation aligns with the traditional understanding of interfacial polymerization, where amine groups tend to exhibit some solubility in the oil phase, leading most amine groups to preferentially insert themselves into the oil phase^{37,38} (³⁷*Langmuir* **2005**, 21, 1884; ³⁸*Science* **2018**, 360, 518). Similar to surfactants, the amphiphilic cage molecules, tend to spontaneously assemble at the oil/water interface (with most amine groups inserted in the oil phase, Supplementary Fig. 6c) due to the interfacial energy considerations.”

4. The membrane is positively charged, since ~4 of 12 amine groups are protonated. However, as shown in the surface zeta potential of the cage composite membrane, the membrane is negatively charged (−3.0 mV) at a neutral pH. A reason should be given.

Reply: We appreciate your observation regarding the surface charge of the membrane and the discrepancy between the expected positive charge and the measured negative zeta potential. Upon further investigation, we have identified the reason for this unexpected negative charge on the top surface of the membrane. This phenomenon can be ascribed to the hydrolysis of the residual acyl chloride groups, leading to the formation of carboxylates (*Nat. Commun.* **2022**, 13, 500; *J. Membr. Sci.* **2011**, 378, 243). This is supported by the presence of 3.2% COO^- , as confirmed through XPS analysis (Supplementary Figure S11, page 17).

However, we acknowledge the validity of your concern and recognize the

importance of addressing this issue. To provide a more comprehensive understanding of the membrane's surface charge, we conducted zeta potential measurements on the back surface of the cage nanofilm using a previously reported sample preparation method (*Adv. Mater.* **2018**, 30, 1705973). The results of these measurements indicated a slight positive charge (1.8 mV, pH~7, Fig. R1) on the back surface of the cage nanofilm. This finding aligns with expectations. The quaternized nitrogen groups face toward the water-phase side during the reaction, leading to their enrichment on the back surface of the cage nanofilm. We hope this explanation clarifies the surface charge characteristics of the membrane and addresses your concern.

Fig. R1 Zeta potential of the top surface and back surface of the networked cage nanofilm.

5. The hydrophobic groups are around the cage window, so how can they reduce the time required for water molecules to traverse the cage and improve the membrane water permeance?

Reply: Your question raises an important point. In many nanofluidic channels, the mass-transfer efficiency is primarily influenced by the interactions between water molecules and the groups surrounding the opening window of the channel (*Science* **2017**, 357, 792). While it might seem counterintuitive that hydrophobic groups around the cage window can enhance water permeability, this phenomenon can be explained

by the nature of these interactions. This effect arises from the lower energy barrier for water molecules to interact with hydrophobic groups (TFSI⁻), compared to strongly hydrophilic groups (Cl⁻). As a result, water molecules can more easily pass through the cage, leading to improved membrane water permeability.

We appreciate your inquiry, and we hope this explanation clarifies how hydrophobic counterions can play a role in enhancing water permeability.

6. There is no discussion about the crystallinity of the prepared POC membranes throughout the manuscript, why?

Reply: We thank the reviewer for pointing out this issue. we have included X-ray diffraction patterns in the Fig. R2, which demonstrate the amorphous nature of the networked cage membranes.

In fact, the crystallinity of organic cages is highly depended on their molecular structure and packing manner. Although imine organic cages with rigid molecular structures have the potential to form highly crystalline packings, they exhibit poor crosslinking reactivity. During a reduction process, the imine groups undergo transformation into more reactive amine groups, resulting in a more flexible molecular skeleton (*J. Am. Chem. Soc.* **2014**, 136, 7583). As a result, it is reasonable to expect the formation of an amorphous nanofilm when using these reduced cages as building blocks for the crosslinking network construction.

Fig. R2 PXR D pattern of crosslinked Cage 1 nanofilm.

Reviewer 2#

In this study, the authors reported an interfacial polymerization technique to fabricate ultrathin membranes from porous cage materials. The researchers showcased the membranes' remarkable water flux. They further demonstrated that by ion exchanging photosensitive ligands, the membrane flux can be regulated through light modulation. I have the following inquiries.

1. The primary concern in this study lies in the underestimated size of the dye molecules. The majority of the dye molecules reported in this work should be approximately 1 nm instead of 0.5 nm. This discrepancy raises significant doubts regarding the actual contribution of the porous cage materials to the membrane's functionality, considering that the window size of the cages is merely 0.5 nm. To address this concern, the authors must provide solid evidence that confirms the transport occurs via the porous cage structure rather than through membrane defects.

Reply: Thanks for your comments. To address your concerns and enhance the clarity regarding the transport mechanism, we have performed additional analyses and experiments.

Firstly, we want to emphasize that the rejection performance of our cage membrane is primarily influenced by the minimum dimension of the dyes, rather than their maximum dimension. It is a common practice in the field of molecular separation membranes to utilize the minimum dimension of molecules as the criterion for determining separation accuracy, as highlighted in recent research (*Nature* **2022**, 609, 58). For instance, Congo red exhibits a minimum dimension of 5.2 Å (as cited from *Nat. Commun.* **2020**, 11, 5882). If our cage membranes possessed significant defects, they would not achieve the high rejection rates, often exceeding 99.0%, as observed in our experiments. We would like to emphasize that, after the filtration of the dye solution, the back surface (facing the permeate side) of the cage composite membrane remains uncolored, as illustrated in Fig. R3a. This observation provides clear evidence that the cage nanofilm is free from defects and exhibits a sufficient density to effectively reject the dye molecules. In contrast, the back surface of the support membranes is prone to

be colored after filtration (Fig. R3b). It's also important to note that we have meticulously addressed the dye absorption effect (no absorption was observed), as detailed in response to comment 5.

Secondly, we conducted experiments on salt rejection to confirm the defect-free feature and achieved a high rejection rate up to 95.0 % for MgSO₄. Further information on these experiments is provided in our response to comment 3.

In addition, we use a fragment of Cage 1 as a monomer to fabricate nanofilm with similar chemical composition but without water channels (*i.e.*, intrinsic pores of cage molecules). This control nanofilm showed an inherent water permeability, P_w , of $1.68 \times 10^{-6} \text{ cm}^2 \text{ s}^{-1}$, which is 6.7 times lower than that of the cage nanofilm, emphasizing the unique role of the cage cavity in water transport.

In summary, we confidently affirm that the ultrahigh water flux exhibited by our cage membrane primarily stems from the water channel effect of the cage molecules, rather than membrane defects.

Fig. R3 Back surfaces of the cage composite membrane (a) and the support membrane (b) after continuous Congo red solution filtration.

2. From Figure 2a, I observed a substantial decrease in water permeance as the reaction time increases or when higher concentrated solutions are utilized. However, in Figure 1h, the membrane thickness does not appear to increase to the same extent. This trend further deepens my concerns regarding the true impact of the porous cage materials on

the membrane's functionality. Therefore, it is essential for the authors to offer a more comprehensive study on the membrane structure.

Reply: We sincerely value your precious comments. To address your concerns and provide further insight into our research, we conducted comprehensive characterizations of the structures of membranes fabricated with varying reaction time and concentrations.

First, all the top surface structures of composite membranes after water filtration test are checked using high-magnification SEM. As shown in Fig. R4, all the top surface of the composite membranes are dense and defect-free.

Then, we provide all the digital images of back surface of composite membrane after dyes filtration (Fig. R5). All the back surfaces are uncolored after Congo red filtration, further confirmed the integrity of the membranes.

Last, we performed XPS tests to analyze the changes in the crosslinking degree of the cage nanofilm concerning variations in cage concentration and reaction time (Fig. R6). The results have elucidated that as the reaction time and cage concentration increased, the crosslinking degree of the network also enhanced. The increase in crosslinking degree could prompt the insertion of more TMC molecules between cage molecules. This may diminish channel connectivity/window accessibility of the porous cage network, ultimately resulting in reduced water permeability. It is worth noting that the accessibility of guests within porous organic cages is highly influenced by the packing model of cage molecules (*Nat. Commun.* **2020**, 11, 4927).

Fig. R4 SEM images of top surface of porous support membrane and composite

membrane fabricated with varying reaction time and concentrations. The top surfaces of composite membranes retained visible substrate profiles, highlighting their ultrathin and flexible characteristics.

Fig. R5 Digital image of cage composite membranes after dyes filtration

Fig. R6 Percentage of crosslinked -NH group and the O/N ratio of the networked cage nanofilm increases with cage concentration and reaction time.

3. In the reported pore size range, i.e. ~ 0.5 nm, the membrane should be able to reject monovalent ions such as Na^+ , or at least reject divalent ions such as Mg^{2+} . The authors should demonstrate the membrane performance in these systems instead of dye molecules.

Reply: We thank the reviewer for pointing out this issue. Our cage membrane has indeed demonstrated salt rejection capabilities, achieving rejection rates of up to 95% (MgSO_4). However, the current work primarily centered on the fabrication of ultrathin networked cage nanofilms, highlighting their distinctive water channel properties and adaptive molecular sieving capabilities. We are actively pursuing the design and development of cage membranes specifically for desalination applications. Thank you for highlighting this aspect, and we are committed to providing comprehensive performance data in future research.

4. Was the reported water permeance measured using pure water or a dye solution? Similarly, regarding the long-term permeation test, is it also conducted with pure water? If the measurements were performed solely with pure water, the long-term test may not offer valuable information on fouling and consequently may have limited usefulness.

Reply: The reported water permeance measurements were indeed conducted using pure water. However, the long-term permeation test involved the use of a dye solution as the feed (Fig. R7). This approach allows us to assess the membrane's performance in more realistic conditions and provides valuable insights into fouling behavior over an extended period. We appreciate your concern, and we have taken these factors into account to ensure the relevance and usefulness of our long-term test data.

Fig. R7 a) cross-flow filtration apparatus and the b) membrane unit used for the continuous filtration test. The feed temperature was maintained at approximately 25°C with the assistance of a water chiller. The cross-flow rate was set at 50 L h⁻¹.

5. Regarding the measurement of dye rejection, although the authors acknowledged the potential adsorption effect by allowing a half-hour waiting period prior to measurement, this duration may be insufficient given the high surface area of the porous cage materials and the relatively low dye concentration. To address this concern adequately, it is necessary for the authors to conduct an adsorption experiment to quantify the adsorption capacity of each dye.

Reply: We appreciate your suggestion, and we have taken action to address this concern. We conducted an adsorption experiment using a Valia-Chien diffusion cell to quantitatively determine the dye adsorption capacity. Membrane samples of cage composite membrane with a diameter of 2 cm were fixed between the diffusion cell. One chamber was filled with a 100 ppm Congo red dye solution, using either water or methanol as the solvent, while the other chamber was filled with pure solvent. The dye concentration in the solution was monitored using UV-vis absorption. As illustrated in Fig. R8 (Also added to Supplementary Information, Supplementary Figure S23, page 24, highlighted with a yellow background), no noticeable variation in UV-vis absorption was observed in the feed chamber, and the permeate chamber remained colorless even after 72 hours. These findings confirm that the membrane did not absorb the dyes.

Related description was also added to the revised manuscript (paragraph 2, Page 8).

“To confirm that dye adsorption did not affect the membrane selectivity, dye adsorption tests using Valia-Chien diffusion cell were performed. No noticeable variation in UV-vis absorption was observed in the feed chamber, and the permeate chamber remained colorless even after 72 hours, indicating that the membrane did not absorb the dyes (Supplementary Fig. 23).”

Fig. R8 Dye adsorption test using a Valia-Chien diffusion cell. Solvent: a) water; b) methanol.

6. The authors utilized a chemistry similar to polyamide membranes in crafting the porous cage membrane, wherein the porous cage molecules served as imine moieties. While the authors claim that the conventional process cannot yield ultrathin membranes,

this assertion is not entirely accurate. It would be advisable for the authors to refer to reference (Science 2015, 348, 1347–1351) to appreciate that ultrathin membranes can indeed be achieved through meticulous process control. It is worth noting that due to the disparity in membrane thickness, the reported performance improvement may be somewhat exaggerated. For instance, although the authors mentioned a 65-fold reduction in permeance for membranes produced through the conventional process, but these membranes are also 10 times thicker. Moreover, an important point omitted by the authors is that membranes produced through the conventional process are much tighter, making them suitable for reverse osmosis (RO) or nanofiltration (NF) applications.

Reply: Thank you for your valuable feedback. In our manuscript, we don't claim that the conventional process is incapable of producing ultrathin membranes. We agree with the reviewer's comments regarding the significant progress made in conventional interfacial polymerization, encompassing both advancements in membrane structure control and industrial applications of RO and NF. The Science paper (*Science* **2015**, 348, 1347) indeed serves as a seminal work in this context and offers valuable insights that have guided our research.

We have underscored the challenge for preparing ultrathin porous organic cage (POC) nanofilm (In the main text, paragraph 2, page 2, highlighted with a yellow background). Our work provides a practical reference for the preparation of chemically linked ultrathin cage nanofilms, a crucial aspect in the development of porous molecule-based materials.

In our study, we sought to ensure a fair and informative comparison with traditional polymeric membranes. To mitigate the influence of thickness, we calculated the intrinsic water permeability (P_w , cm^2s^{-1}) of the networked cage nanofilms using a well-established method (*J. Membr. Sci.* **2019**, 590, 117297) (In the Supplementary Information, section 2.3.2 Water transport calculation, page 6). This approach helps provide a more equitable basis for evaluating the performance of our membranes.

7. Standalone membranes are generally not preferred in practical applications due to their limited mechanical strength. Therefore, I suggest that the authors consider employing a similar process as that used for reverse osmosis (RO) or nanofiltration membranes to create a thin film on a porous support. If such an approach is pursued and the membrane performance exhibits a comparable improvement, it would greatly enhance my confidence in supporting this work.

Reply: We appreciate your suggestion regarding the fabrication of membranes on porous supports. In our study, we opted to fabricate cage nanofilms at the free oil/water interface. This approach was chosen for its convenience in exploring the intricate structure-function relationship, as demonstrated in previous research (*Adv. Mater.* **2018**, 30, 1705973).

We recognize the convenience of directly fabricating nanofilms on porous supports. Accordingly, we employed the traditional interfacial polymerization method, using Cage 1 and its fragment (R,R-1,2-diaminocyclohexane) as aqueous monomers, respectively. Notably, a porous support membrane with a CNT interlayer was utilized, following the established interlayer strategy known for forming uniform nanofilms (*Science* **2015**, 348, 1347; *Chem. Soc. Rev.* **2021**, 50, 6290). As displayed in Fig. R9, a dense and smooth cage nanofilm was successfully formed on the CNT interlayer. The resulting membrane exhibited a comparable water permeance ($318.5 \text{ L m}^{-2}\text{h}^{-1}\text{bar}^{-1}$) with excellent dye rejection (Congo red, 99.2 %), a performance 50 times higher than that of the membrane prepared with the fragment (Fig. R9). However, it is crucial to acknowledge that achieving such ultrathin layers directly on porous support may necessitate innovative support membrane design and further optimization. We are committed to conducting a comprehensive investigation in future research.

Fig. R9 Fabrication of cage composite membrane using a traditional interfacial polymerization method. Aqueous monomer concentration: 1 mM; TMC concentration: 6 mM; Aqueous immersing time: 5 min; Air-drying time: 5min; Reaction time: 2 min.

8. The authors stated that the cage molecules are aligned at the interface. However, it is crucial for them to present comprehensive characterization data to substantiate this claim.

Reply: This appears to be a misunderstanding. We have not stated the alignment of cage at the interface, as it remains a big challenge to probe the alignment of molecules at an oil/water interface. As mentioned in our previous response, much like surfactants, amphiphilic cage molecules naturally tend to self-assemble at the oil/water interface driven by interfacial energy considerations. To support this claim, we have conducted dynamic simulations, which provide further insights into the molecular arrangements at the interface. For additional details, please refer to the response to comment 3 from Reviewer 1, where we elaborate on our simulation findings.

9. The authors solely focused on investigating water permeation in their study. However, it would be highly beneficial to extend their investigation to include other commonly used solvents such as methanol, ethanol, NMP, hexane, and so on. By studying the

permeation behavior of these additional solvents, a more comprehensive understanding of the membrane's performance can be gained.

Reply: We appreciate your constructive suggestion. While our primary focus was on elucidating water transport behaviors within the nanofluidic channels of the cage membranes, we acknowledge the importance of evaluating the membrane's performance in the context of organic solvent nanofiltration.

To provide a more comprehensive understanding of the membrane's separation behavior, we conducted an assessment of membrane permeation using Congo red as the probe molecule and various typical solvents. Note that all the membranes used to test the solvent permeance were subjected to a Congo red rejection test to confirm their integrity. The summarized results of these experiments are presented in Table R1, indicating the remarkable solvent permeance of the cage composite membrane.

We believe this expanded investigation will contribute to a more thorough assessment of the membrane's capabilities in various practical applications. Your valuable suggestion has played a crucial role in enriching the scope of our research.

Solvent	Permeance (L m ⁻² h ⁻¹ bar ⁻¹)
Water	360.0
methanol	444.3
ethanol	167.2
N,N-Dimethylformamide	258.3
acetonitrile	764.0
acetone	573.2

Fig. R10 Conge red rejection test after solvent permeance determination. The permeate is colorless and dye molecule is undetectable with UV-vis absorption.

Reviewer 3#

This is an interesting manuscript based on the use of cages for the preparation of thin film composite membranes.

1. My first concern is why the authors do not mention previous manuscripts using porous organic cages for the preparation of thin film composite membranes using interfacial polymerization. Examples are Li et al. *Nature Comm.* 14, 3112 (2023) with a quite similar approach or He et al. *Nature Mater.* 21, 463 (2022). As for light-sensitive membranes with azo segments, the authors probably know the paper of Liu et al. *Science Adv.* 6, eabb3188, 2020.

Reply: Thank you for bringing this to our attention. We have taken your feedback into consideration, and we would like to clarify our approach to referencing previous research. While we did compare our cage membrane with the work from He Ai et al. (*Nat. Mater.* 2022, 21, 463) in the main text (paragraph 2, page 10, highlighted with a yellow background), we regretfully missed mentioning the manuscript by Li et al. (*Nat. Commun.* 2023, 14, 3112), as it became available online after we had completed our manuscript. We have now incorporated the paper by Li et al. as a reference (Reference 20) in our revised manuscript to acknowledge its relevance.

Additionally, we appreciate your reference to the paper by Liu et al. (*Sci. Adv.* 2020, 6, eabb3188) concerning light-sensitive membranes with azo segments. We confirm that this paper has already been cited in our manuscript (Reference 52).

Your valuable feedback helps ensure that our research is well-contextualized and that we appropriately acknowledge the relevant contributions of previous work in the field.

2. I do not understand Fig 2f. What is the x-axis?

Reply: We appreciate your inquiry regarding Figure 2f. In this figure, we indeed compared the water permeability between cage composite membranes and conventional polymeric membranes. However, we realize that we did not provide a specific label for the x-axis.

To clarify, the x-axis in Figure 2f represents different membranes that used for the water permeability comparison. We apologize for the lack of labeling, and we have included this information in our revised manuscript (Fig. 2f, page 9) for better clarity and understanding.

3. Fig. 2c: molecular rejection of which molecule size?

Reply: We appreciate your question, and we apologize for any ambiguity. In Fig. 2c, the molecular rejection data pertains specifically to the rejection of Congo red molecules. To provide clearer context, we have added this information to the revised manuscript (In the main text, title of Fig. 2, page 9) to ensure a more comprehensive understanding of the data presented.

“c) Aqueous nanofiltration performance comparison of cage composite membranes with commercial and literature reported membranes. The molecule rejection data refer to selectivity of Congo red.”

4. Fig. 2b: what is the cut off in g mol^{-1} ?

Reply: we want to emphasize that the rejection performance of our cage membrane is primarily influenced by the minimum dimension of the dyes, rather than their molecule weigh. The cut off dimension is around 5.2 \AA .

5. More information on the set-up used for the UV permeance experiments is needed.

Reply: We appreciate your feedback regarding the UV permeance experiments. To provide more clarity and transparency, we have included an image of the experimental setup (as shown in Fig. R11) in the revised supplementary information (Scheme S1, Section 2.3.4, page 7).

Fig. R11 The set-up used for the light controlled graded molecular separation. Prior to first-round filtration, the membrane unit underwent UV irradiation, causing the cage nanofilm to transition into a cis-state. Subsequently, the first-round filtration was conducted, resulting in the detection of only 4-nitrophenol in the permeate, with Congo red and methyl orange being nearly completely rejected. Following this, the membrane unit underwent visible (Vis) irradiation, inducing a transition of the cage nanofilm into a trans-state. As a result, methyl orange became permeable, while Congo red continued to be rejected. After flushing the residual methyl orange from the feed with an excess of deionized water, the pure phase of Congo red could be obtained.

REVIEWERS' COMMENTS

Reviewer #1 (Remarks to the Author):

My concerns regarding the pore size, interfacial energy, surface charge, channel hydrophobicity and the crystallinity of the membrane have been well addressed. The revised manuscript can be accepted for publication.

Reviewer #2 (Remarks to the Author):

The authors have provided convincing responses to all my questions, especially the new adsorption data in Fig. S8. I support the acceptance of this work at the current state.

Reviewer #3 (Remarks to the Author):

The authors revised the manuscript and I recommend the publication.

Response to Reviewer

Reviewer 1#

My concerns regarding the pore size, interfacial energy, surface charge, channel hydrophobicity and the crystallinity of the membrane have been well addressed. The revised manuscript can be accepted for publication.

Reply: We thank the reviewer for the reviewing of our manuscript.

Reviewer 2#

The authors have provided convincing responses to all my questions, especially the new adsorption data in Fig. S8. I support the acceptance of this work at the current state.

Reply: We are grateful for your constructive comments and suggestions to improve the quality of our work.

Reviewer 3#

The authors revised the manuscript and I recommend the publication.

Reply: We thank the reviewer for the reviewing of our manuscript again.